# From NAFLD to Chronic Liver Diseases. Assessment of Liver Fibrosis through Non-Invasive Methods before Liver Transplantation: Can We Rely on Them?

**Pasquale Auricchio** [1] and **Michele Finotti** [2,3,*]

1    DISCOG, University of Padua, 35182 Padua, Italy
2    Annette C. and Harold C. Simmons Transplant Institute, Baylor University Medical Center, Dallas, TX 75204, USA
3    4th Surgery Unit, Regional Hospital Treviso, DISCOG, University of Padua, 31100 Padua, Italy
*    Correspondence: mi6le@libero.it

**Abstract:** Chronic liver injury and subsequent liver fibrosis are usually a slow process without any specific or no clinical signs, resulting in pathological conditions with a poor chance of improvement through medical and surgical treatment, which if not promptly recognized, often lead to a liver transplant as the only therapeutic option. On the other hand, screening and follow-up are hard to establish in large populations using regularly invasive methods such as biopsies and other expensive diagnostic tools due to cost and a lack of adequate specificity and sensibility. In the last few years, a large variety of serological and radiological tests have been proposed to assess liver fibrosis. In this review, we will consider the most commonly used scores to evaluate liver fibrosis, with a special focus on the NAFLD pathogenesis. We will try to answer the question: can we rely on them?

**Keywords:** liver fibrosis; NAFLD; cirrhosis; non-invasive methods; liver transplantation

## 1. Introduction

Liver fibrosis has to be suspected in almost all patients with chronic liver injury, secondary to the healing response to chronic liver damage (due to alcohol abuse, metabolic syndrome, chronic hepatitis, etc.) [1]. Nonetheless, a critical aspect of the consequences of liver injury is its timing, because for acute lesions such as fulminant hepatitis, the healing process does not involve scar development, despite a conspicuous amount of fibrogenic stimuli, as described by Bataller et al. [2], which is the typical aspect in chronic liver injury, resulting in a disruption of the liver architecture. In particular, the increasing development of the extracellular matrix (ECM) that is associated with uneven production and clearance are the main features of chronic liver disease, leading to liver fibrosis and stiffness [3].

A fibrotic liver presents damages that are independent of the underlying cause (from alcohol to viral infection), and these may be localized differently in the liver, not only according to the pathogenesis, but also to patients [4]. However, chronic liver disease and fibrosis are not easily detectable in daily practice, especially in the primary care setting, due to a lack of accessible diagnostic tools, and it is difficult to identify patients that will ultimately need a transplant evaluation. Current liver tests, such as serum aminotransferases, have low sensibility and specificity for identifying fibrosis. As has been stated in many guidelines, liver biopsy is considered as the gold standard for the assessment of a large variety of liver diseases, including liver fibrosis. However, in the very first stages of liver fibrosis—which could be months to years of ongoing injury—it is possible that it is not cost-effective, or that the procedure carries some risks. In addition, a liver biopsy is not feasible as a screening test [5]. Understanding the burden of the problem is necessary for evaluating the prevalence of fibrosis around the world. In Europe, non-alcoholic fatty liver disease (NAFLD) is considered as the main cause of liver fibrosis

leading to end-stage liver disease and liver transplantation, with a prevalence of fibrosis of 0.7% and 7.5% in population-based cohorts, and 18–27% in cohorts with risk factors for chronic hepatic injury [6]. In Asia and North America, in less than 30 years, the prevalence of NAFLD increased from 20% to 30%, with an estimated prevalence of liver fibrosis in 3.2% and 10.3% of the patients, reaching 17% in some Asiatic studies [7,8].

In this picture, an evaluation system that is able to predict the development, progression, and degree of chronic liver disease and liver fibrosis would be extremely beneficial, especially to evaluate the possible patients needing a specialized evaluation and possible transplant evaluation [9].

In a large span of chronic liver diseases, NAFLD is the major cause, affecting 25% of the world's adult population [10,11] according to different studies [12]: up to 20% of patients affected by NAFLD develop non-alcoholic steatohepatitis (NASH) [13], with the chances to progress in cirrhosis and hepatocellular carcinoma arising [14–17]. Moreover, NAFLD has a heavy impact in terms of reduced quality of life and pressure on the healthcare system, and it is an increasing indication of liver transplantation [18–23].

Considering the NAFLD as the hepatic component of the metabolic disease [24–26], the strong correlation with type 2 diabetes mellitus (T2DM) is higher than in the general population [27–30], and since it is also strongly associated with obesity [24], the urge toward large screening in the era of the global obesity epidemic is mandatory. Recently, a new terminology was introduced, namely, metabolic dysfunction-associated fatty liver disease (MAFLD). In MAFLD, hepatic steatosis is associated with obesity, T2DM, and/or metabolic dysfunction, regardless of alcohol intake [31].

A proper assessment should combine accuracy and reproducibility, and most of all, it should be responsive to changes in fibrosis levels over time. This could be interesting, especially in the setting of the recent multiple ongoing trials on NAFLD evaluating the effects of medications in reducing the progression or even reversal of liver fibrosis. This challenge has been embraced through different methods, with a view to a progressive picture from fibrosis to cirrhosis. We discuss in this paper the major groups:

- Blood-based biomarkers score,
- Elastrography,
- Combined methods.

This review aims to summarize the recent evidence about these topics, the reporting methods, and studies, to specifically assess liver fibrosis secondary to NAFLD chronic liver damage before/during the liver transplant evaluation.

## 2. A Blood-Based Biomarkers Score

Scores with blood-based biomarkers were designed to obtain tools that are usable in everyday practice. We can divide those tools into two major groups: simple and complex biomarkers blood tests. The difference between them lies in the parameters used in routine blood tests versus specific markers (e.g., serum tissues metalloproteinases, hyaluronate).

Fibrogenesis, the principal element of chronic liver disease and liver fibrosis/stiffness, is secondary to excessive ECM formation. Biomarkers that are able to reflect the hepatic ECM changes in chronic liver injury may be used to assess liver fibrosis development, progression, and regression.

The combination of regular indirect markers of liver fibrosis and clinical parameters drove the development of tests such as FIB-4, APRI, and NFS (Table 1). Key points regarding the reliabilities of those scores are the AUROC (area under the receiver operating characteristics), sensibility, and sensitivity.

**Table 1.** Comparison between simple biomarkers tests.

| | Variables | Formula | Cutoffs in NAFLD | |
|---|---|---|---|---|
| | | | **Low Risk** | **High Risk** |
| FIB4 | Platelet count, age, AST, ALT | AQR X AST)/(Platelets × (sgr(ALI)) | <1.30 (NPV = 90%) | 11.30–3.25 > 2.67 (PPV = 80%) |
| APRI | AST, platelet count | I (AST/ULN AST) × 1001/platelets (10/L) | <1.0 (NPV = 84%) | >1.0 (PPV = 37%) |
| NFS | Age, BMI, hyperglycemia, platelet count, albumin, AST/ALT ratio | $-1.675 + 0.037 \times$ age (y) $+ 0.094 \times$ BMI (kg/m$^2$) $+ 1.13 \times$ IFG/ diabetes (yes 1, no = 0) $+ 0.99 \times$ AST/ALT ratio $- 0.013$ platelet $\times 10°$/L) $- 0.66 \times$ albumin (g/dL) | <−1.455 (NPV = 88%) | −1.455–0.675 >0.675 (PPV = 82%) |

- Fibrosis score 4 (FIB-4). Sterling et al. [32] developed a score using age, ALT, AST, and platelet count to estimate the amount of scarring in the liver. Based on a retrospective analysis, liver histology was performed in 832 HIV/HCV-coinfected patients. Liver fibrosis was assessed via the Ishak score, and multivariate logistic regression analysis revealed that platelet count (PLT), age, AST, and INR were significantly associated with fibrosis. After an additional analysis without INR, an alternative model with PLT, age, AST, and ALT was developed. According to these data, (FIB-4) was designed: age ([yr] AST [U/L])/((PLT [109/L]) (ALT [U/L])1/2). The AUROC of the index was 0.765 for differentiation between Ishak stages 0–3 and 4–6. At a cutoff of <1.45 in the validation set, the negative predictive value to exclude advanced fibrosis (stage 4–6) was 90%, with a sensitivity of 70%. A cutoff of >3.25 had a positive predictive value of 65% and a specificity of 97%. Using these cutoffs, 87% of the 198 patients with FIB-4 values outside 1.45–3.25 would be correctly classified, and liver biopsy could be avoided in 71% of the validation group. The authors argued that these individuals could potentially have avoided liver biopsy, with an overall accuracy of 86%.
- Ast–Platelet Ratio Index (APRI). Using a retrospective cohort study involving 270 patients with chronic hepatitis C, Chun-Tao Wai et al. [33] developed a score with only AST and platelets as the variables. Those patients underwent a liver biopsy over 25 months and were divided into a training set (n192) and a validation set (n78). To amplify the opposing effects of liver fibrosis on AST and platelet count, the AUCs of APRIs for predicting significant fibrosis and cirrhosis were 0.80 and 0.89, respectively, in the training set. Using optimized cut-off values, significant fibrosis could be predicted accurately in 51%, and cirrhosis in 81% of patients. The AUCs of APRI for predicting significant fibrosis and cirrhosis in the validation set were 0.88 and 0.94, respectively. A meta-analysis of 40 studies showed that an APRI score of greater than 1.0 provides a sensitivity of 76% and a specificity of 72% for predicting cirrhosis; meanwhile, an APRI score of greater than 0.7 reaches a sensitivity of 77% and a specificity of 72% for predicting fibrosis [34]. Meta-regression analysis indicated that the APRI accuracies for both significant fibrosis and cirrhosis were affected by histological classification systems, but they were not influenced by the interval between biopsy and APRI, or blind biopsy.
- NFS. For this score, Angulo et al. [35] used age, hyperglycemia, body mass index, platelet count, albumin, and the AST/ALT ratio as independent indicators of advanced liver fibrosis. A total of 733 patients with NAFLD confirmed via liver biopsy were divided into two groups to construct (n = 480) and validate (n = 253) the scoring system. The formula results are more greatly articulated compared to others' scores: 1.675 1 0.037 age (y) 1 0.094 BMI (kg/m$^2$) 1 1.13 IFG/diabetes (yes 5 1, no 5 0) 1 0.99 AST/ALT ratio–0.013 platelet (109/L)–0.66 albumin (g/dL). A scoring system with these six variables had an AUROC curve of 0.88 and 0.82 in the estimation and

validation groups, respectively. By applying the low cutoff score (1.455), advanced fibrosis could be excluded with high accuracy (negative predictive values of 93% and 88% in the estimation and validation groups, respectively). By applying the high cutoff score (0.676), the presence of advanced fibrosis could be diagnosed with high accuracy (positive predictive values of 90% and 82% in the estimation and validation groups, respectively). By applying this model, a liver biopsy would have been avoided in 549 (75%) of the 733 patients, with correct prediction in 496 (90%). Despite the enrollments of many patients, these were included from different centers in the world that have a particular interest in studying NAFLD, and thus, some referral bias could not be ruled out.

Many studies have evaluated the roles of these three indexes (FIB-4, NFS, and APRI) in the specific setting of NAFLD.

Vilar-Gomez et al. evaluated them in >18-year-old patients with biopsy-proven NASH planning for lifestyle intervention (diet and physical activity changes). A second biopsy was performed in the follow-up to assess fibrosis and to use it to evaluate the index reliability. The study showed that among the NFS, APRI, and FIB-4 scores, NFS was the score that can predict better an improvement or worsening fibrosis. To note, the authors also proposed a scoring system with the variables being able to predict fibrosis (change from baseline in platelets count, HbA1c, and the normalization of ALT), with an AUC of 0.96 (95% CI, 0.94–0.99) [36].

The relationships among FIB-4, NFS, and APRI in NAFLD and fibrosis were also evaluated and confirmed by Siddiqui et al. [37]. Recently APRI, FIB-4, and NFS were validated in a population with a high prevalence of NAFLD. FIB-4 showed the highest AUROC, with a cut-off$\geq$ 1.3 being a possible indicator of liver fibrosis [38]. As described, studies reported inconsistent results regarding the best indicator to predict fibrosis in patients with NAFLD. A recent systemic review showed that FIB-4, NF,S, and APRI are good as liver biopsies to stratify the NAFLD patient morbidity and mortality risk, but more refined models are needed to predict the change in fibrosis status [39].

Blood tests involving complex biomarkers such as serum tissue metalloproteinases and hyaluronic acid could directly describe fibrogenesis and fibrinolysis, but these require specialist laboratory assessment. The Enhanced Liver Fibrosis Score (ELF), Hepascore, Fibrospect, II, and ADAPT algorithms may be more accurate than simple scores, but they could also imply higher costs (Table 2).

**Table 2.** Comparison between complex biomarkers test.

| | | | Accuracy in Fibrosis | |
| | Variables | Formula | F1–F2 | F3–F4 |
|---|---|---|---|---|
| ELF | TIMP-1, PIIINP, HA | $2.494 + 0.846 \ln(\text{CHA}) + 0.735 \ln(\text{CPIIINP}) + 0.391 \ln(\text{CTIMP-1})$ | Cutoff PIIINP = 242.3 ng/mL Sensitivity 73.8% Specificity 90% | Cutoff PIIINP = 698.7 ng/mL Sensitivity 75% Specificity 96% |
| Hepascore | Age, sex, total bilirubin, GGT, alpha-2-macroglobulin, HA | $Y = \text{EXP}(-4.185818 - (0.0249 \times \text{age}) + (0.7464 \times \text{sex}) + (1.0039 \times \text{A2M}) + (0.0302 \times \text{HA}) + (0.0691 \times \text{Bil-t}) - (0.0012 \times \text{GGT})$ | | Cutoff > 0.55 Sensitivity 82% Specificity 65% |
| Fibrospect II | HA, TIMP-1, alpha-2-macroglobulin | Patented Formula | | Sensitivity 71.8% Specificity 73.9% |
| ADAPT | Age, DM2, platelet count PRO-C3 | $\exp(\log_{10}((\text{age} \times \text{PROC3})/\sqrt{\text{platelet count}}))$ | | Sensitivity 81% Specificity 73% |

- Enhanced Liver Fibrosis Score (ELF). Developed by the European Liver Fibrosis Group, the Enhanced Liver Fibrosis (ELF) score provides a single value using an

algorithm combining the quantitative serum measurements of tissue inhibitor of metalloproteinases-1 (TIMP-1), amino-terminal propeptide of type III procollagen (PIIINP), and hyaluronic acid (HA). To calculate the ELF score, the following equation was originally designed: ELF score = 2.494 + 0.846 ln (CHA) + 0.735 ln (CPIIINP) + 0.391 ln (CTIMP-1). After a validation study on a cohort of NAFLD patients, the test accuracy was not reduced by excluding age as a parameter in the algorithm [40]. The results obtained through this validation cohort showed excellent performance in distinguishing advanced fibrosis in patients with NAFLD, with an AUROC of 0.90. Anyway, the test performance resulted in less accuracy in patients with chronic hepatitis C, because of parameters such as age and gender [41]. The same problem has been shown for the evaluation of chronic hepatitis B, with an AUROC of 0.67 [42]. Regarding the specifics, biomarkers are necessary to point out the roles of TIMP-1, PIIINP, and HA in the development of liver fibrosis: tissue inhibitor of metalloproteinases 1 (TIMP-1) drives the remodeling process in the liver via matrix metalloproteases (MMPs) [43]; meanwhile, serum PIIINP and HA were positively correlated with early liver fibrosis stage ($r = 0.622$, $p < 0.001$, and $r = 0.41$, $p < 0.001$, respectively). Through a receiver operating curve (ROC) analysis, it has been shown that serum PIIINP was the most effective for the diagnosis of fibrosis grade among the other markers used for this score. The areas under the ROC curves (AUROCs) for serum PIIINP for diagnosing fibrosis stages $\geq$F1, $\geq$F2, $\geq$F3, and F4 (cirrhosis) were 0.843, 0.789, 0.82, and 0.891, respectively. The cut-off serum PIIINP value for predicting fibrosis stage $\geq$F1 was 242.3 ng/mL, with 73.8% sensitivity and 90% specificity. The cut-off value for predicting cirrhosis was 698.7 ng/mL, with 75% sensitivity and 96% specificity [44]. The NICE guidelines suggested ELF as "the most cost-effective and the most appropriate test for advanced fibrosis in adults with NAFLD". However, a recent meta-analysis evaluated the role of ELF, considering 14 different studies in NAFLD patients, with liver biopsy being used as a reference standard, showing that ELF has a high sensitivity but a limited specificity to exclude fibrosis, especially in cases of low disease prevalence [45].

- Hepascore. This score consists of a correlation between non-specific (age, sex, total bilirubin, and GGT) and specific markers of fibrosis, such as alpha-2-macroglobulin and hyaluronic acid levels [46]. Adam et al. designed this complex equation in two steps: Y = EXP ($-4.185818 - (0.0249 \times$ age) + ($0.7464 \times$ sex) + ($1.0039 \times$ A2M) + ($0.0302 \times$ HA) + ($0.0691 \times$ Bil-t) $- (0.0012 \times$ GGT). After obtaining Y, the simplified Hepascore formula results in Y = Y/(1 + Y). At values that are less than or equal to 0.2, the negative predictive value to exclude fibrosis is 98%. At values that are greater than or equal to 0.8, the positive predictive value for predicting cirrhosis is 62%. Therefore, this score offers a good negative predictive value and could be reliable for excluding significant fibrosis, but it is not so effective in predicting cirrhosis: more parameters for such a prediction are mandatory. Hepascore could predict significant fibrosis (F2–4), as proven by the AUROC in validation sets (0.81). A cutoff score of >0.55 was best for predicting significant fibrosis, with sensitivities and specificities of 82% and 65%, respectively, and positive and negative predictive values of 70% and 78%. Up-to-date studies with a high level of evidence, evaluating the role of Hepascore in predicting fibrosis in the specific setting of the NAFLD population, are still lacking.

- Fibrospect II. FIBROSpect II is a predictive algorithm for fibrosis stages F2 to F4, which combines hyaluronic acid, tissue inhibitor of a metalloproteinase-1 (TIMP-1), and alpha-2-macroglobulin. As has been already seen for other scores describing F2–F4 fibrosis, specific markers such as hyaluronic acid (HA), TIMP-1, and alpha2-macroglobulin (A2M) could offer predictive accuracy, achieving an AUROC of 0.831. At an index cut-off of 0.36 and a prevalence for F2–F4 of 52%, the results in all 696 patients indicated positive and negative predictive values of 74.3 and 75.8%, respectively, with an accuracy of 75%, and the reliability of this score is mostly focused on moderate–severe fibrosis. An index score of greater than 0.42 correlates with the presence of stages F2 to F4 fibrosis. Based on data from the test manufacturer involving

696 persons with chronic HCV infection, the overall sensitivity at this cutoff is 80.6%, and the specificity is 71.4% [47]. Fibrospect carries the same issues appearing with the previously described scores: good results in terms of excluding the presence of liver fibrosis, but a poor ability to describe its progression [48]. A recent study validated TIMP-1, A2M, and HA in NAFLD patients, showing a great ability to predict mild to advanced fibrosis (a ROC curve of 0.856, a sensitivity of 79.7%/specificity of 75.7%), which is superior compared to FIB-4 and NFS [49].

- ADAPT score: A marker of type III collagen formation has been recently associated with fibrosis development in patients with chronic hepatitis C: the so-called PRO-C3. Daniels et al. recently evaluated its role in NAFLD patients, validating the ADAPT score. The PRO-C3 was measured with an enzyme-linked immunosorbent assay (ELISA) in a total of 431 patients with biopsy-proven NAFLD: 150 patients in the derivation and 281 in the validation cohort. The first result was that PRO-C3 is also strongly related to fibrosis in NAFLD patients. In the derivation cohort, patients with advanced fibrosis (F $\geq$ 3) had a high level of PRO-C3 compared to the mild/moderate group ($p < 0.0001$). In addition, PRO-C3 is associated with the severity of the disease and the stage of fibrosis, with an additional ability to correlate with hepatocyte ballooning, lobular inflammation, and steatosis. The ADAPT score, considering age, the presence of diabetes, PRO-C3 (a marker of type III collagen formation), and platelet count, was then created, with an AUROC of 0.86 (95% CI 0.79 to 0.91) in the derivation and 0.87 in the validation cohort (95% CI 0.83 to 0.91) for advanced fibrosis. Furthermore, the authors showed the superiority of the ADAPT score compared to APRI, FIB-,4, and the NAFLD fibrosis score (NFS) to predict fibrosis in NAFLD patients [50]. Recently, Tang et al. evaluated the ADAPT score in an Asian cohort, including 851 biopsy-proven MAFLD patients. The ADAPT score showed an AUROC of 0.865, with a better ability to predict fibrosis compared to PRO-C3 alone or other non-invasive fibrosis tests (APRI score, Fibrosis-4, BARD, and NAFLD fibrosis score) [51].

## 3. Elastography

The usage of elastography lies in the increasing stiffness occurring through liver fibrosis development. The main methods to evaluate it are transient elastography, shear wave elastography, and magnetic resonance elastography [52–54].

Transient Elastography (Fibroscan). Fibroscan is the combination of ultrasound (US) (5 MHz) and low-frequency (50 Hz) elastic waves obtaining a one-dimensional (1D) transient elastography. Originally used for chronic hepatitis C, this technique achieves a quantification of liver fibrosis, with a good level of reproducibility. Liver elasticity measurements are reproducible (standardized coefficient of variation: 3%), operator-independent, and well correlated (partial correlation coefficient = 0.71, $p < 0.0001$) to fibrosis grade (METAVIR) (Figure 1). The AUROC curves obtained from Sandrin et al. [55] were 0.88 and 0.99 for the diagnosis of patients with significant fibrosis ($\geq$ F2) and with cirrhosis (F4), respectively. Therefore, its results indicate it to be a reliable, reproducible, and widely used method for assessing liver fibrosis.

Shear Wave Elastography (SWE). Based on B-mode ultrasound, shear wave elastography can estimate hepatic fibrosis without invasive measures. It consists of a real-time image evaluation with B-mode ultrasound: liver stiffness could be obtained after anatomical information and the variation of color images, which describe also liver homogeneity [56]. It does not offer the same results as Fibroscan because of the lack of reproducibility. This is why it is less commonly used in clinical assessments.

| Metavir Scoring System for Fibrosis staging | |
|---|---|
| F0 | No Fibrosis can be detected |
| F1 | Fibrosis exists with expansions of portal zones |
| F2 | Fibrosis exists with expansions of most portal zones and occasional bridging |
| F3 | Fibrosis exists with expansion of most portal zones, marked bridging, and occasional nodules |
| F4 | Presence of cirrhosis |

**Figure 1.** Metavir Scoring System for Fibrosis staging. Metavir is an acronym for "meta-analysis of histological data in viral hepatitis". Confusingly, there is also a web-based viral genome analysis platform called metavir, which has nothing to do with the scoring system.

Magnetic Resonance Elastography (MRE). Magnetic resonance is largely used in liver diagnosis because it offers a reproducible, non-invasive, and quantitative assessment of fibrosis. Yin et al. [57] studied the sensitivity and specificity of the technique in diagnosing liver fibrosis, to reduce the number of patients requiring a biopsy. Comparing 35 normal volunteers and 50 patients with chronic liver disease, the measurements of hepatic fat-to-water ratios were obtained to assess the potential for fat infiltration to affect the stiffness-based detection of fibrosis, which resulted in an increase according to the arising stage of fibrosis. Receiver operating curve analysis showed that with a shear stiffness cut-off value of 2.93 kilopascals, the predicted sensitivity and specificity for detecting all grades of liver fibrosis are 98% and 99%, respectively. Receiver operating curve analysis also provided evidence that can discriminate between patients with moderate and severe fibrosis (grades 2–4), and those with mild fibrosis (sensitivity, 86%; specificity, 85%). Hepatic stiffness does not appear to be influenced by the degree of steatosis.

## 4. Combined Methods

To achieve a higher level of accuracy in determining and characterizing the different levels of liver fibrosis, blood markers, scores, and instrumental methodology has been combined, with the major purpose of simulating the results obtained with a liver biopsy. The most commonly used and described methods are the Fibrometer and the SAFE score.

Fibrometer. The main purpose of Calès et al. [58] in the development of the Fibrometer was to describe the fibrotic development in viral and alcoholic chronic liver diseases. The authors decided to use 51 different blood markers, combined with Fibrotest, Fibrospect, ELFG, APRI, and FORNS. In an explorative cohort involving 383 patients affected by viral hepatitis, the fibrotic area through image analyses was at first assessed; thereafter, a Metavir staging (F2–F4) was used to standardize the evaluation, combined with platelets, prothrombin index, aspartate aminotransferase, 2-macroglobulin, hyaluronate, urea, and age. The area under the receiver operating characteristic showed a good discriminatory ability within the stages F2–F4, achieving a value of 0.883. A successive validation with 120 patients improved the AUROC by 0.892. To subcategorize and improve the accuracy of this methodology, two main groups were analyzed: patients affected by alcoholic liver disease were tested with a combination of prothrombin index, A2M, hyaluronate, and platelets; patients affected by viral hepatitis were tested for hyaluronate, glutamyltransferase, bilirubin, platelets, and apolipoprotein A1. Since the first introduction of the Fibrometer, other versions have been developed to stratify a larger amount of liver diseases, such as FibroMeter VCTE, to detect advanced fibrosis (sensitivity: 83.5% (95%CI 0.58–0.94); specificity: 91.1% (95%CI 0.89–0.93)), followed by FibroMeter V2G (sensitivity: 83.1% (95%CI 0.73–0.90); specificity: 84.4% (95%CI 0.62–0.95)) and FibroMeter NAFLD (sensitivity: 71.7% (95%CI

0.63–0.79); specificity: 82.8% (95%CI 0.71–0.91)). In a recent review and meta-analysis [59], FibroMeter tests showed an acceptable level of sensitivity and specificity in detecting advanced fibrosis, and also in patients with NAFLD, but further comparisons need to be collected.

Sequential Algorithm for Fibrosis Evaluation (SAFE algorithm). The SAFE algorithm is the combination of the APRI score, Fibrotest, and Metavir staging (F2–F4) [60]. A multicentric study involving 2035 patients affected by the hepatitis C virus [61] compared the accuracy of SAFE versus histology. With an AUROC of 0.92 for cirrhosis and 0.89 for fibrosis, the Sequential Algorithm for Fibrosis Evaluation also achieved a reduction in liver biopsies of up to 46,5%. The performance of the score was mostly influenced by age and BMI, but with an adjustment of Fibrotest-Fibrosure cutoffs, it could be improved. Nonetheless, a SAFE Algorithm does not apply to NAFLD evaluation [62]. The accuracy of this algorithm has been proven in further studies for the hepatitis C population [63].

## 5. Discussion

Detecting liver fibrosis in the general population would be essential for recognizing the population suitable for possible treatments, reducing the patient's chance to progress to end-stage liver disease and the need for liver transplantation. This process would ultimately reduce the burden on the waiting list for a liver transplant, but the fibrosis screening in the population with only non-invasive scores is a matter of debate.

In the NAFLD population, for example [64], Dobbie et al. showed that FIB-4, APRI, and ELF tend to underestimate the presence of liver fibrosis. Recently, Nielsen et al. compared the abilities of PRO C-3 and ADAPT, FIB-4, and APRI to predict liver fibrosis and NASH in the context of CENTAUR screening (NCT02217475). The study confirmed the ability of PRO-C3 to correlate with NASH and fibrosis severity, and the superiority of the ADAPT score compared to FIB-4 and APRI to detect liver fibrosis and NASH. The ADAPT score performed better than other scores because it includes a direct marker of fibrogenesis (PRO-C3) associated with the main risk factors of advanced fibrosis in NAFLD patients: advanced age, diabetes, and low platelet count. Eslam et al. evaluated in three independent cohorts the association of the ADAPT score and liver stiffness measurements in evaluating liver fibrosis. The study confirmed the superiority of the ADAPT score compared to other scores (aspartate aminotransferase-to-platelet ratio index, fibrosis-4, BARD, and non-alcoholic fatty liver disease fibrosis score), and the addition of liver stiffness measurements leads to a diagnostic accuracy of 92.5%, with 98% and 100% negative predictive value, and the ability to exclude advanced fibrosis in low-risk populations. This tool may help the physician to stratify the population needing specialistic evaluation and liver biopsy [65].

Park [66] proposed to detect at first the patients that would benefit the most from screening for hepatic fibrosis, since patients with type 2 diabetes had a lower performance in non-invasive tests [67]. Despite those results, other studies [68] showed a satisfying accuracy by these methods in detecting advanced fibrosis in patients with NAFLD.

Recently, Corey et al., using aptamer-based proteomics, was able to measure multiple blood proteins, and identified two proteins (the ADAMTSL2 protein and an 8-protein) as a possible biomarker of significant fibrosis in the NASH population. In particular, the 8-protein panel can distinguish NAFL/NASH fibrosis stage 0–1 from fibrosis stage 2–4 with an AUROC of 0.87–0.89, and the ADAMTSL2 protein alone can distinguish NAFL/NASH fibrosis stage 0–1 from fibrosis stage 2–4 with an AUROC of 0.83–0.86. Furthermore, the 8-protein panel and ADAMTSL2 protein had superior performance to the NAFLD fibrosis score and fibrosis-4 score [69].

Ishiba et al. compared the diagnostic accuracy of fibrosis via FIB-4, NFS, APRI, and type IV collagen 7S (COL4-7S) in patients with NAFL, with and without diabetes. The AUROC of COL4-7S was significantly larger than those of the other scores in patients with NAFLD with diabetes than in those without diabetes, concluding that the COL4-7S measurement might be the best tool for evaluating advanced fibrosis in NAFLD, especially in patients with diabetes [70].

Better performances were reported by Xiao et al. [71] using MRE and SWE to diagnose and offer aging in NAFLD groups. The authors added that among non-invasive blood scores, FIB-4 and NFS performed better than the other scores described, but the results were still not comparable to elastography. In a comparison between MRE and Fibroscan in the bariatric population [72], Garteiser et al. showed that MRE is superior to transient elastography in detecting NAFLD, which was thereafter confirmed through liver biopsy.

A combination of several tools and techniques seems to be the best way to predict liver fibrosis, especially in the NAFLD population. Recently Cassinotto et al. evaluated in 577 consecutive patients the diagnostic performance and need for liver biopsy in unclassified patients for the diagnosis of advanced fibrosis (F ≥ 3) of a two-step strategy (laboratory test plus transient elastography VCTE or SWE) and a three-step strategy (laboratory tests and two elastography methods). The authors concluded that SWE is comparable to VCTE to evaluate liver fibrosis in NAFLD; furthermore, a three-step strategy, maintaining great accuracy, reduced the need for a liver biopsy [73].

Mozes et al. performed a meta-analysis considering 37 studies involving 5735 patients, aiming to evaluate the diagnostic performance of the current most common tools (VCTE, FIB-4, and NFS), and their ability to reduce the need for liver biopsy. The study showed that the sequential combination of FIB-4 followed by VCTE had a sensitivity and specificity (95% CI) of 66% (63–68) and 86% (84–87), with 33% needing a biopsy to establish a final diagnosis. If FIB-4 was followed by LSM, the resulting sensitivities and specificities were 38% (37–39) and 90% (89–91), respectively, with 19% needing biopsy, with a specific cut-off. The authors showed and confirmed that the two-step algorithms are better than one, even if the cut-off of the single test is still a matter of debate [74,75].

If we switch the focus to other chronic liver diseases such as primary sclerosing cholangitis and autoimmune hepatitis [76,77], we can appreciate that the trend that by now has been described: MRE is a very effective method in detecting F2 fibrosis and subsequent development.

It has to be remarked that as previously described, non-invasive blood tests have been designed for patients affected by viral hepatitis or alcoholic hepatitis; therefore, it could be reasonable to think that a general use of scores may not be so accurate for all liver diseases, and specifically, NAFLD.

Furthermore, these scores have to be considered in real clinical practice. Most of the tests reported can be expensive, increasing the cost and making it difficult to use them in the current clinical setting. It is mandatory to have more evidence and data on these tests in the context of NAFLD, to apply them with efficacy, and ideally, at a lower possible cost. In the recent AASLD practice guidance on the clinical assessment and management of non-alcoholic fatty liver disease, they stated that "Although FIB-4 is statistically inferior to other serum-based fibrosis markers such as the ELF panel, FIBROSpect II, and imaging-based elastography methods to detect advanced fibrosis FIB-4 is still recommended as a first-line assessment for general practitioners and endocrinologists based on its simplicity and minimal cost" [78].

On the same line, the Japanese Society of Gastroenterology and the Japanese Society of Hepatology proposed an affordable clinical application of these scores in NAFLD patients: hepatic fibrosis markers such as HA and COL4-7S, and a scoring system such as the FIB-4 index and NFS should be used as the first screening. In the case of a high suspicion of liver fibrosis based on the previous tests, a gastroenterology/hepatology consultation should be proposed for further evaluation (VCTE, MRE, or liver biopsy) [79].

To note, recent data are emerging about evaluating NAFLD recurrence after liver transplant, and the same scores used in the general population may be helpful for following up on this specific topic. To date, compared to LT for alcohol liver disease or other indications [80], the reduction in the NAFLD risk factors is not mandatory before LT, with possible disease recurrence after the surgery. Furthermore, the overall survival after LT is increasing, exposing the patients to NAFLD risk factors, and these are worsened by long-term immunosuppression, leading to possible de novo NAFLD after LT, regardless of

the initial indication for LT. In particular, NAFLD and NASH have been reported to recur in 62% and 33% of patients, respectively, and de novo NAFLD/NASH after LT are reported in 20%/10% of the patients [81,82]. Studies have shown that the same tests proposed to identify NAFLD/NASH and fibrosis in non-LT patients may be applied in the transplanted population, such as transient elastography [83,84]. Recently, Alhinai et al. evaluated the role of the controlled attenuation parameter (CAP) and the serum biomarker cytokeratin 18 (CK-18) to investigate NAFLD/NASH in transplanted patients. The authors showed that, compared to liver histology, CAP has a 76% accuracy for diagnosing NAFLD, while the accuracy of CAP plus CK-18 to diagnose NASH is 82%. In this setting, new tools and the validation of those available to monitor NAFLD/NASH recurrence and fibrosis after LT are mandatory.

## 6. Conclusions

Non-invasive methods are feasible and offer a large variety of solutions to all physicians who have to handle patients that are at risk of liver fibrosis because of chronic liver diseases. Despite the different approaches and their combinations, it is questionable to assume that non-invasive methods are as good as liver biopsy, which so far remains the gold standard for the diagnosis and staging of fibrosis, especially in the context of liver transplant evaluation.

In terms of screening and the early detection of liver fibrosis, blood-based markers were less efficient; meanwhile, good results could be obtained with elastography.

The next challenge for non-invasive methods should be to obtain direct markers for NAFLD that are as good as those obtained for viral hepatitis and alcoholic hepatitis, and that are also applicable in the follow-up of possible NAFLD recurrence in the transplanted population.

**Author Contributions:** Conceptualization, methodology, validation, writing—original draft preparation, writing—review and editing, P.A. and M.F. All authors have read and agreed to the published version of the manuscript.

**Funding:** This research received no external funding.

**Institutional Review Board Statement:** Not applicable.

**Informed Consent Statement:** Not applicable.

**Data Availability Statement:** Not applicable.

**Conflicts of Interest:** The authors declare no conflict of interest.

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
