# Peer review of "From NAFLD to Chronic Liver Diseases. Assessment of Liver Fibrosis through Non-Invasive Methods before Liver Transplantation: Can We Rely on Them?"

_2673-3943, doi:10.3390/transplantology4020008_

Round 1

Reviewer 1 Report

Excellent review and accurate portrayal of important concept we are dealing with.  The paper was easy to read and covers the different aspects of non-invasive techniques to assess liver fibrosis.  

Author Response

Dear reviewer, thank you for your time and comments. 

Reviewer 2 Report

The manuscript is a mere summary of the available biochemical and instrumental test without a clear emphasis on NAFLD/NASH.  Most of the references in section 2 and 3 are addressed on populations different from NAFLD like HCV, HBV and coinfected HIV-HCV patients. However, in both of the tables, are reported specifical cut off values for NAFLD not mentioned in the previous text. This is really confusing and as a consequence the reader looses its focus on NAFLD.

In the original form, the manuscript doesn't add anything to what is already known.

I strongly suggest to revise the paper addressing specifically NAFLD population and choosing some particular fields (i.e. application of non invasive evaluation of fibrosis in prediction of prognosis, liver related events and development of HCC) offering a critical review of the benefits and limits  of these tools.

Author Response

Dear Reviewer,

Thank you for your comments and suggestions. We extensively reviewed the paper based on your indications, updating the bibliography, and citing papers more pertinent to NAFLD patients. Also, we evaluated the role of these scores and tools in the current clinical practice.

In particular:

  • At the end of the Blood-based Biomarkers score section we added the following part: “…Many studies evaluated the role of these three indexes (FIB-4, NFS, and APRI) in the specific setting of NAFLD. Vilar-Gomez et al. evaluated them in >18-year-old patients with biopsy-proven NASH planning for lifestyle intervention (diet and physical activity changes). A second biopsy was performed in the follow-up to assess fibrosis and use it to evaluate the index reliability. The study showed that among NFS, APRI, and FIB-4 scores, NFS was the score that can predict better improvement or worsening fibrosis. To note, the authors also proposed a scoring system with the variables able to predict fibrosis (change from baseline in platelets count, HbA1c, and normalization of ALT) with an AUC of 0.96 (95% CI, 0.94-0.99). [36] The relationship among FIB-4, NFS, and APRI in NAFLD and fibrosis was also evaluated and confirmed by Siddiqui et al. [37]. Recently APRI, FIB-4, and NFS were validated in a population with a high prevalence of NAFLD. FIB-4 showed the highest AUROC, with a cut-off≥ 1.3 as a possible indicator of liver fibrosis [38]. As described, studies reported inconsistent results regarding the best indicator to predict fibrosis in patients with NAFLD. A recent systemic review showed that FIB‐4, NF,S, and APRI are good as liver biopsies to stratify the NAFLD patient morbidity and mortality risk, but more refined models are needed to predict change in fibrosis status. [39]
  • On line 392 we added “…NICE guidelines suggested ELF as “the most cost-effective and the most appropriate test for advanced fibrosis in adults with NAFLD”. However, a recent meta-analysis evaluated the role of ELF considering 14 different studies in NAFLD patients, with liver biopsy used as a reference standard, showing that ELF has high sensitivity but limited specificity to exclude fibrosis, especially in cases of low disease prevalence. [45]”
  • Line 468: “…A recent study validated TIMP-1, A2M, and HA in NAFLD patients, showing a great ability to predict mild to advanced fibrosis (ROC curve of 0.856, sensitivity 79.7%/specificity of 75.7%), superior compared to FIB-4 and NFS [49]
  • Line 489: “…Recently, Tang et al. evaluated the ADAPT score in an Asian cohort including 851 biopsy-proven MAFLD patients. The ADAPT score showed an AUROC of 0.865, with a better ability to predict fibrosis compared to PRO-C3 alone or other non-invasive fibrosis tests (APRI score, Fibrosis-4, BARD, and NAFLD fibrosis score) [51].
  • We modified the discussion evaluating the current role of these scores/tools and the current guidelines. In particular: “…It has to be remarked that as previously described, non-invasive blood tests have been designed for patients affected by viral hepatitis or alcoholic hepatitis, therefore could be reasonable to think that general use of scores could be not so accurate as for all liver diseases, and specifically NAFLD. Furthermore, these scores have to be considered in real clinical practice. Most of the tests reported can be expensive, increasing the cost and making it difficult to use them in the current clinical setting. Is mandatory to have more evidence and data on these tests in the context of NAFLD, to apply them with efficacy and ideally at lower cost possible. In the recent AASLD practice guidance on the clinical assessment and management of nonalcoholic fatty liver disease, they stated that “Although FIB-4 is statistically inferior to other serum-based fibrosis markers such as the ELF panel, FIBROSpect II, and imaging-based elastography methods to detect advanced fibrosis FIB-4 is still recommended as a first-line assessment for general practitioners and endocrinologists based on its simplicity and minimal cost" [78] On the same line, the Japanese Society of Gastroenterology and the Japanese Society of Hepatology proposed an affordable clinical application of these scores in NAFLD patients: hepatic fibrosis markers, such as HA, COL4-7S and a scoring system like the FIB-4 index and NFS should be used as first screening. In case of a high suspect of liver fibrosis based on the previous tests, a gastroenterology/hepatology consultation should be proposed for further evaluation (VCTE, MRE, or liver biopsy) [79]

Reviewer 3 Report

The review is written well and interesting for the readers.

Author Response

Dear Reviewer, 

Thank you for your time and comments.

Best regards

Round 2

Reviewer 2 Report

Thanks for your extensive revision of the manuscript

Please in Tab 1 respect the order of the list reported in the text 

Paragraph 3 "Elastography":  citing Hepatic Ultrasound scan is a kind of impurity. My advice is to remove this subparagraph

Author Response

Dear reviewer, 

Thank you for your suggestions. As you indicate, we changed the order in table 1 and delete the subparagraph about the Hepatic Ultrasound. 

Best regards,